# Survival Benefit of Hepatic Arterial Infusion Chemotherapy over Sorafenib in the Treatment of Locally Progressed Hepatocellular Carcinoma

**DOI:** 10.3390/cancers13040646

**Published:** 2021-02-05

**Authors:** Hideki Iwamoto, Takashi Niizeki, Hiroaki Nagamatsu, Kazuomi Ueshima, Takako Nomura, Teiji Kuzuya, Kazuhiro Kasai, Yohei Kooka, Atsushi Hiraoka, Rie Sugimoto, Takehiro Yonezawa, Akio Ishihara, Akihiro Deguchi, Hirotaka Arai, Shigeo Shimose, Tomotake Shirono, Masahito Nakano, Shusuke Okamura, Yu Noda, Naoki Kamachi, Miwa Sakai, Hiroyuki Suzuki, Hajime Aino, Norito Matsukuma, Satoru Matsugaki, Kei Ogata, Yoichi Yano, Takato Ueno, Masahiko Kajiwara, Satoshi Itano, Kunitaka Fukuizumi, Hiroshi Kawano, Kazunori Noguchi, Masatoshi Tanaka, Taizo Yamaguchi, Ryoko Kuromatsu, Atsushi Kawaguchi, Hironori Koga, Takuji Torimura

**Affiliations:** 1Division of Gastroenterology, Department of Medicine, Kurume University School of Medicine, Kurume 830-0011, Japan; iwamoto_hideki@med.kurume-u.ac.jp (H.I.); shimose_shigeo@med.kurume-u.ac.jp (S.S.); shirono_tomotake@med.kurume-u.ac.jp (T.S.); nakano_masahito@med.kurume-u.ac.jp (M.N.); okamura_shyuusuke@kurume-u.ac.jp (S.O.); noda_yuu@med.kurume-u.ac.jp (Y.N.); kamachi_naoki@med.kurume-u.ac.jp (N.K.); sakai_miwa@med.kurume-u.ac.jp (M.S.); suzuki_hiroyuki@med.kurume-u.ac.jp (H.S.); ryoko@med.kurume-u.ac.jp (R.K.); tori@med.kurume-u.ac.jp (T.T.); 2Iwamoto Internal Medicine Clinic, Kitakyusyu 802-0832, Japan; iwamotos@orion.ocn.ne.jp; 3Department of Gastroenterology, Juntendo University, Bunkyo-ku 113-8421, Japan; h-nagamatsu@juntendo.ac.jp; 4Department of Gastroenterology and Hepatology, Faculty of Medicine, Kindai University, Osaka 589-8511, Japan; kaz-ues@med.kindai.ac.jp; 5Department of Gastroenterology and Neurology, Faculty of Medicine, Kagawa University, Miki 761-0793, Japan; nomura-takako@takamatsu.jrc.or.jp; 6Department of Gastroenterology and Hepatology, Nagoya University Graduate School of Medicine, Nagoya 466-8560, Japan; teiji.kuzuya@fujita-hu.ac.jp; 7Division of Gastroenterology, IMS Sapporo Digestive Disease Center General Hospital, Sapporo 063-0842, Japan; kaz-k@yc4.so-net.ne.jp; 8Division of Hepatology, Department of Internal Medicine, Iwate Medical University School of Medicine, Iwate 028-3695, Japan; ykooka@iwate-ed.ac.jp; 9Gastroenterology Center, Ehime Prefectural Central Hospital, Matsuyama 790-0024, Japan; hirage@m.ehime-u.ac.jp; 10Department of Hepato-Biliary-Pancreatology, National Hospital Organization Kyushu Cancer Center, Fukuoka 811-1395, Japan; sugimoto.rie.jw@mail.hosp.go.jp; 11Department of Gastroenterology, Hachinohe Red Cross Hospital, Aomori 039-1104, Japan; takiyama@iwate-med.ac.jp; 12Department of Gastroenterology, Osaka National Hospital, Osaka 540-0006, Japan; ishihara.akio.mq@mail.hosp.go.jp; 13Department of Gastroenterology, Kagawa Rosai Hospital, Marugame 763-8502, Japan; akihiro41@me.com; 14Department of Gastroenterology, Maebashi Red Cross Hospital, Maebashi 371-0811, Japan; h-arai@maebashi.jrc.or.jp; 15Division of Gastroenterology, Department of Medicine, Social Insurance Tagawa Hospital, Tagawa 826-0023, Japan; aino0217@kph.biglobe.ne.jp; 16Department of Gastroenterology, Kurume General Hospital, Kurume 830-8521, Japan; matsukuma-norito@kurume.jcho.go.jp; 17Department of Gastroenterology, Tobata Kyoritsu Hospital, Kitakyusyu 804-0093, Japan; k-ikyoku@kyoaikai.com; 18Department of Gastroenterology, Kurume University Medical Center, Kurume 839-0863, Japan; keiogata@med.kurume-u.ac.jp; 19Department of Gastroenterology, Saga Central Hospital, Saga 849-8522, Japan; syano@kumin.ne.jp; 20Department of Gastroenterology, Asakura Medical Association Hospital, Asakura 838-0069, Japan; ueno.tk@asakura-med.or.jp; 21Department of Gastroenterology, Chikugo City Hospital, Chikugo 833-0041, Japan; mkajiwara1@mac.com; 22Department of Gastroenterology, Kurume Central Hospital, Kurume 830-0001, Japan; angiolion@h9.dion.ne.jp; 23Department of Gastroenterology, National Hospital Organization Kyushu Medical Center, Fukuoka 810-8563, Japan; fukuizumi.kunitaka.zu@mail.hosp.go.jp; 24Department of Gastroenterology, St. Mary’s Hospital, Kurume 830-8543, Japan; kawano_hiroshi@kurume-u.ac.jp; 25Department of Gastroenterology, Omuta City Hospital, Omuta 836-0861, Japan; hisyo@ghp.omuta.fukuoka.jp; 26Department of Gastroenterology, Yokokura Hospital, Miyama 839-0215, Japan; mazzo6528@me.com; 27Center for Comprehensive Community Medicine Faculty of Medicine, Saga University, Saga 840-0027, Japan; akawa@cc.saga-u.ac.jp

**Keywords:** hepatocellular carcinoma, intra-arterial infusions, sorafenib, molecular targeted therapy

## Abstract

**Simple Summary:**

Not all patients with hepatocellular carcinoma (HCC) benefit from treatment with systemic treatments such as sorafenib. Hepatic arterial infusion chemotherapy (HAIC) is the treatment using an indwelling catheter port system. The regimen of HAIC used in the study is New FP which is consisted of a fine-powder cisplatin and 5-fluorouracil. In the study, for the patients with locally progressed HCC, such as HCC with vascular invasion, initial administration of local hepatic treatment using HAIC was superior to systemic treatment using sorafenib.

**Abstract:**

BACKROUND: Not all patients with hepatocellular carcinoma (HCC) benefit from treatment with molecular targeted agents such as sorafenib. We investigated whether New-FP (fine-powder cisplatin and 5-fluorouracil), a hepatic arterial infusion chemotherapy regimen, is more favorable than sorafenib as an initial treatment for locally progressed HCC. METHODS: To avoid selection bias, we corrected the data from different facilities that did or did not perform New-FP therapy. In total, 1709 consecutive patients with HCC initially treated with New-FP or sorafenib; 1624 (New-FP, *n* = 644; sorafenib *n* = 980) were assessed. After propensity score matching (PSM), overall survival (OS) and prognostic factors were assessed (*n* = 344 each). Additionally, the patients were categorized into four groups: cohort-1 [(without macrovascular invasion (MVI) and extrahepatic spread (EHS)], cohort-2 (with MVI), cohort-3 (with EHS), and cohort-4 (with MVI and EHS) to clarify the efficacy of each treatment. RESULTS: New-FP prolonged OS than sorafenib after PSM (New-FP, 12 months; sorafenib, 7.9 months; *p* < 0.001). Sorafenib treatment, and severe MVI and EHS were poor prognostic factors. In the subgroup analyses, the OS was significantly longer the New-FP group in cohort-2. CONCLUSIONS: Local treatment using New-FP is a potentially superior initial treatment compared with sorafenib as a multidisciplinary treatment in locally progressed HCC without EHS.

## 1. Introduction

Macroscopic vascular invasion (MVI) and extrahepatic spread (EHS) are two of the factors that define “advanced” stages of hepatocellular carcinoma (HCC). Currently, molecular targeted agents (MTAs), including sorafenib, are the standard treatment for advanced HCC [1,2,3,4]. Although the approved MTAs prolong survival even in patients with advanced HCC, including HCC with MVI (MVI-HCC), their therapeutic effects are unsatisfactory because the basal prognosis of patients with advanced HCC is poor [5,6]. Therefore, further research is needed in advanced HCC treatment.

Hepatic lesion progression directly correlates with prognosis in patients with HCC, even in patients with HCC with EHS [7]. Thus, hepatic lesion control can improve prognosis. Hepatic arterial infusion chemotherapy (HAIC) is a frequently reported local treatment for MVI-HCC [8,9,10,11,12] that directly and consecutively delivers anti-cancer drugs into HCC in the liver. HAIC can theoretically increase local anti-cancer drug concentrations in the liver and reduce systemic adverse events due to anti-cancer drugs. According to previous reports about HAIC for HCC, stronger local control is an important feature of HAIC compared with systemic chemotherapy [13,14]. However, there is no consensus that HAIC is beneficial for advanced HCC because few clinical trials of HAIC for HCC have been conducted. Recently, evidence to support the efficacy of HAIC in the treatment of HCC has been increasing [15,16,17]. We previously reported a regimen of HAIC named “New-FP (FP: CDDP and 5-FU)” in the treatment of advanced HCC [18,19,20]. New-FP is a regimen that comprises fine-powder cisplatin (DDP-H) suspended in lipiodol (an oil-based radio-opaque contrast agent) and 5-fluorouracil (5-FU). Suspended DDP-H-lipiodol is expected to have an enhanced permeability and retention effect, and lipiodol works as a carrier of the drug delivery system [21]. We have accumulated clinical evidence regarding New-FP compared with sorafenib by reporting single- and multicenter retrospective studies and a non-randomized prospective multicenter study [18,19,20]. In our previous study, New-FP significantly prolonged the survival of patients with MVI-HCC compared with sorafenib (median survival time [MST]: New-FP 30.4 months, sorafenib 13.2 months (*p* < 0.01)). However, this study had a small sample size (*n* = 64) [20]. Here, we compared patients’ overall survival (OS) with advanced HCC treated in facilities that offer New-FP treatment and those that do not offer New-FP treatment using propensity score matching (PSM). We also conducted subgroup analyses to clarify the usefulness of each treatment for various tumor conditions. We aimed to show a proof-of-concept that initial treatment with local hepatic treatment using HAIC provides better OS over sorafenib in patients with locally progressed HCC.

## 2. Materials and Methods

### 2.1. Study Design

This was a multicenter retrospective cohort study of patients with HCC. We included patients with HCC who were initially treated with New-FP or sorafenib from the Kurume Liver Cancer Study group and eight other hospitals between March 2009 and June 2019.

The following inclusion criteria were used: (i) HCC diagnosed by biopsy or radiological evaluation using enhanced computed tomography (CT) or magnetic resonance imaging (MRI) combined with serum tests for tumor markers; (ii) age >18 years; and (iii) complete follow-up from the initial treatment of this study until death or the study censor time (30 June 2019). The patients who were initially treated with sorafenib were excluded from the New-FP group. This study protocol was approved by the Ethics Committee of the Kurume University School of Medicine (No. 19004) and was conducted according to the Helsinki Declaration. The requirement for written informed consent was waived because of the retrospective study design.

### 2.2. Patients

Patient information, including sex, age, HCC etiology, Child-Pugh (C-P) class, alpha-fetoprotein (AFP), des-γ-carboxy prothrombin (DCP), tumor size, and the presence of MVI and EHS were collected from medical records. HCC was classified using the Barcelona Clinic Liver Cancer (BCLC) staging system [22]. Additionally, MVI and EHS were diagnosed using enhanced CT and MRI, respectively. Severe MVI was defined as tumor invasion into the first branch or trunk of the portal vein.

#### 2.2.1. Evaluation Items

The following items were evaluated: (i) OS before PSM, (ii) OS after PSM, (iii) factors associated with poor prognosis after PSM, and (iv) which treatment to administer initially under various tumor conditions using subgroup analysis. Subgroup analyses were conducted using the data before PSM to clarify the role of New-FP and sorafenib under various tumor conditions: cohort-1, without MVI and EHS; cohort-2, with MVI and without EHS; cohort-3, without MVI and with EHS; and cohort-4, with MVI and EHS (Figure 1).

#### 2.2.2. Refinement to Avoid Bias

Here, several refinements were conducted to avoid bias in the study. (1) The sorafenib group’s data were collected from the hospital that do not offer New-FP therapy to avoid selection bias. (2) The two groups’ data were collected from the same period to avoid treatment bias. (3) The patients who were initially treated with sorafenib were excluded from the New-FP group. (4) The collected data were not analyzed by any person who administered New-FP or sorafenib. The data were analyzed by an independent statistician to avoid bias. (5) The data were performed using PSM.

#### 2.2.3. PSM

To overcome different distributions of covariates among the New-FP and sorafenib groups, PSM was performed according to the methods previously reported [23]. The propensity score was estimated using a logistic regression model with the following variables: sex, age, HCC etiology, C-P class, tumor size, presence of MVI, presence of EHS, AFP level, and DCP level. A one-to-one nearest-neighbor matching algorithm with an optimal caliper of 0.3 without replacement was used to generate 344 pairs of patients. As *p*-values could be biased by population size, the PSM results were also reported as effect size: <0.2 indicated a negligible difference, <0.5 indicated a small difference, <0.8 indicated a moderate difference, and other values indicated a large difference. The c-statistic was 0.86. Thus, 688 patients with HCC (New-FP (*n* = 344) and sorafenib (*n* = 344); 1:1 matching) were analyzed.

### 2.3. Treatment Protocol

#### 2.3.1. Sorafenib

Sorafenib (Nexval; Bayel Co., Ltd., Osaka, Japan) was administered orally to patients with HCC according to industry recommendations. Adverse events (AEs) were assessed using the National Cancer Institute Common Terminology Criteria for Adverse Events, version 4.0. In patients who were diagnosed with progression after sorafenib administration, second-line treatments, including transcatheter arterial chemoembolization (TACE) and HAIC, except for New-FP, lenvatinib, regorafenib, ramucirumab, or investigational drugs, were allowed.

#### 2.3.2. New-FP

All patients in the New-FP group initially underwent catheter implantation as described below. No patients in the New-FP group had received sorafenib previously. The detailed regimen of New-FP is described below. In patients who were diagnosed with progression after New-FP administration, second-line treatments, including TACE and HAIC, except New-FP, sorafenib, lenvatinib, regorafenib, ramucirumab, or investigational drugs, were allowed.

#### 2.3.3. Catheter Implantation Procedure for HAIC

All HAIC treatments were conducted after the insertion of an implanted catheter (Appendix A, Piolax, Boston, MA, USA) indwelled from the right femoral artery. The indwelling method of the implanted catheter was selected from the gastroduodenal artery (GDA) coiling method (Appendix A), peripheral hepatic artery fixation method (Appendix A), and coaxial method (Appendix A), as required. The GDA, right gastric artery, posterior superior pancreaticoduodenal artery, and accessory left gastric artery were occluded using metallic coils (Piolax) to avoid gastroduodenal ulcers or pancreatitis in cases where the anti-cancer drugs were distributed into these arteries. Finally, the port was subcutaneously implanted into the front femoral region (Sofa Port, Nipro Pharma Corporation, Osaka, Japan).

#### 2.3.4. HAIC Regimen: New-FP

The regimen of New-FP is shown in Appendix A. New-FP comprises the fine-powder CDDP (DDP-H, IA-Call, Nippon Kayaku, Tokyo, Japan) suspended with ethiodized oil (Lipiodol; Guerbet, Villepinte, France) and 5-FU. For the inpatient regimen of New-FP, 50 mg of DDP-H was suspended in 5–10 mL of lipiodol, of which the amount was determined by tumor volume. On day-1, DDP-H-lipiodol suspension was injected from the implanted catheter under angiography, followed by the injection of 250 mg of 5-FU. Then, 1,250 mg of 5-FU was continuously injected using an infusion balloon pump for 5 days (SUREFUSER PUMP, Nipro Pharma Corporation, Osaka, Japan). This regimen was applied once a week for the first 2 or 3 weeks. For the outpatient regimen of New-FP, 20–30 mg of DDP-H-lipiodol suspension was injected under angiography, followed by the injection of 1,250 mg of 5-FU using an infusion balloon pump for 5 days every 2 weeks. The New-FP regimen was administered depending on the time course of tumor progression, as required. These regimens were continued until the appearance of severe AEs or tumor progression. The time course of the representative CT images in New-FP therapy is shown in Appendix A.

### 2.4. Statistical Analyses

All data are expressed as numbers or medians (ranges). All statistical analyses were performed using JMP statistical analysis software (JMP Pro version 14, SAS Institute Inc., Cary, NC, USA). The OS was calculated using the Kaplan-Meier method and analyzed using the log-rank test. Factors associated with prognosis were evaluated using univariate and multivariate analyses with the Cox proportional hazards model, which derived p-values, hazard ratios, and their 95% confidence intervals. Variables were entered into the multivariate regression model using the stepwise procedure based on the Akaike information criteria (AIC), that is, the selected model had the minimum AIC value among candidate models. A two-tailed *p*-value < 0.05 was considered statistically significant.

## 3. Conclusions

### 3.1. Patient and Tumor Characteristics

The study flow diagram is shown in Figure 2. A total of 1709 consecutive patients diagnosed with HCC, including 671 patients in the New-FP group and 1038 patients in the sorafenib group, were registered. Among them, 10 patients were excluded because sorafenib was initially performed in the New-FP group. Moreover, 75 patients were excluded from the analysis because of incomplete data. A total of 1624 patients, including 644 patients in the New-FP group and 980 patients in the sorafenib group, were enrolled for the PSM analysis. The median follow-up period was 15.8 months.

Patients’ and tumors’ characteristics before and after PSM are summarized in Table 1. Before PSM, 644 and 980 patients were enrolled in the New-FP and sorafenib groups, respectively. The mean age of patients in the New-FP and sorafenib groups was 67.9±11.02 and 70.11±9.52 years, respectively (*p* < 0.001). The number of hepatitis C virus-related HCC cases was significantly higher in the sorafenib group (*p* < 0.001). The New-FP group had significantly fewer C-P class A cases and significantly more C-P class C cases *(p* < 0.001). However, the sorafenib group had significantly more BCLC stage B/C cases (*p* < 0.001). The average maximum tumor size was significantly larger in the New-FP group (*p* < 0.001). There were significantly more patients with MVI-HCC in the New-FP group *(p* < 0.001). Among them, there were significantly more patients with severe MVI-HCC portal vein invasion into the first branch or trunk in the New-FP group (*p* < 0.001). However, the number of patients with EHS-HCC was significantly lower in the New-FP group than in the sorafenib group (*p* < 0.001). The median AFP and DCP levels were not significantly different in the two groups (*p* = 0.430 and *p* = 0.655, respectively). After the PSM, 344 patients were extracted in both groups. There were no significant differences in any of the variables after PSM (Table 1).

### 3.2. OS Curves before PSM

The OS was significantly longer in the New-FP than in the sorafenib group in the pre-PSM data (Figure 3A, hazard ratio (HR) 1.2 (95% confidence interval: 1.1–1.4), *p* < 0.001). The MSTs of the New-FP and sorafenib groups were 12 and 11 months, respectively.

### 3.3. OS Curves after PSM

OS was significantly better in the New-FP than in the sorafenib group after PSM (Figure 3B, HR 1.5 [1.3–1.8], *p* < 0.001). The MSTs of the New-FP and sorafenib groups were 12 and 7.9 months, respectively. Even after overcoming different distributions of covariates among the New-FP and sorafenib groups due to PSM, the New-FP group had significantly prolonged survival than the sorafenib group.

### 3.4. Prognostic Analysis for Survival before and after PSM

Multivariate analyses for OS were performed using the data before and after PSM with the following items: treatment group; sex; age; etiology; tumor size; MVI, severe MVI, and EHS presence; C-P class; AFP; and DCP. Amonth these all factors, sorafenib treatment, the presence of EHS, severe MVI and Child-Pugh Class B were independent factors associated with poor prognosis in the data before PSM (Appendix A). Among these all factors, Sorafenib treatment, the presence of EHS and severe MVI were independent factors associated with poor prognosis in the data after PSM (Table 2).

### 3.5. Subgroup Analysis in the Four Cohort Groups

To clarify the role of New-FP and sorafenib under various tumor conditions, subgroup analyses were conducted in four cohort groups which were classified according to the presence of MVI and/or EHS (Figure 1) of Each cohort’s survival curve is shown in Figure 4.

### 3.6. Cohort-1: Without MVI and EHS

Patient and tumor characteristics in cohort-1 are shown in Appendix A. The patients in the New-FP and sorafenib groups were 73 and 335, respectively. Age, etiology, tumor diameter, and C-P class were significantly different in the two groups. The MST of the New-FP and sorafenib groups was 18 and 15 months, respectively (Figure 4A); however, the difference was not significant. Multivariate analysis revealed that sorafenib treatment and a larger tumor diameter (>95.8 mm) were independent factors that influenced poor prognosis in cohort-1 (Appendix A).

### 3.7. Cohort-2: With MVI and without EHS

Patient and tumor characteristics in cohort-2 are shown in Appendix A. The patients in the New-FP and sorafenib groups were 442 and 149, respectively. Only the C-P class was significantly different in the two groups. The MSTs of the New-FP and sorafenib groups were 13 and 8 months, respectively (Figure 4B). The New-FP group showed significantly longer survival than the sorafenib group (*p* < 0.001). Multivariate analysis revealed that the sorafenib group, C-P class B and presence of severe MVI were independent factors that influenced poor prognosis in cohort-2 (Appendix A). In cohort-2, we also performed PSM and compared MST between two groups (Appendix A). The MSTs of the New-FP and sorafenib groups were 15 and 7.9 months, respectively (*p* < 0.001).

### 3.8. Cohort-3: Without MVI and with EHS

Patient and tumor characteristics in cohort-3 are shown in Appendix A. The patients in the New-FP and sorafenib groups were 13 and 356, respectively. Age was significantly different between the two groups (*p* = 0.032). Tumor diameter was significantly larger in the New-FP group (*p* = 0.007). There was no significant difference between the two groups in hepatitis B virus etiology. The MSTs of the New-FP and sorafenib groups were 6 and 11 months, respectively (Figure 4C). The sorafenib group had significantly prolonged survival compared with the New FP group. Multivariate analysis revealed that a larger tumor diameter (>65.2 mm) and C-P class B were independent factors that influenced poor prognosis (Appendix A). Hepatitis B virus etiology was a better prognostic factor (Appendix A).

### 3.9. Cohort-4: With MVI and EHS

Patient and tumor characteristics in cohort-4 are shown in Appendix A. The patients in the New-FP and sorafenib groups were 116 and 140, respectively. Only the C-P class was significantly different between the two groups (*p* = 0.001). The MSTs of the New-FP and sorafenib groups were 7 and 5 months, respectively (Figure 4D). There was no significant difference in OS between the two groups. Multivariate analysis revealed that the presence of severe MVI was an independent poor prognostic factor (Appendix A). In cohort-4, we could perform PSM and compared the MST between the two groups (Appendix A). The MSTs of the New-FP and sorafenib groups were 8 and 5 months (*p* = 0.089).

## 4. Discussion

This is the first study that compared a single regimen of HAIC (New-FP) with MTA (sorafenib) in patients with HCC using PSM and a large sample size. After PSM, the New-FP group had prolonged OS compared with the sorafenib group. Sorafenib treatment and the presence of EHS and severe MVI, were independent factors associated with poor prognosis in these groups. Subgroup analyses revealed that New-FP was the most effective in the group with MVI and without EHS. The MST of sorafenib was superior to that of New-FP in cohort-3 (without MVI and with EHS). This study provides proof-of-concept that local treatment using New-FP is a potentially superior initial treatment compared with sorafenib, as a multidisciplinary treatment for locally progressed HCC without EHS.

Currently, MTAs are the standard treatment for TACE-refractory HCC and advanced HCC according to worldwide treatment guidelines [24,25,26,27]. However, not all patients with HCC benefit from treatment with MTA. Specifically, locally progressed HCC such as MVI is one of the poorest therapeutic factors in the treatment of MTAs [28,29]. Therefore, various local therapeutic modalities including TACE, transcathter arterial radioembilization and HAIC, have advanced in treatment of HCC. HAIC is a traditional treatment for HCC [8,30,31,32]. Many studies have reported the effectiveness of HAIC for HCC [8,13,33,34]; however, most of them were retrospective, had a small population, and were single-arm. Recently, several randomized prospective studies of HAIC for HCC treatment have been conducted. He et al. reported that the FOLFOX-HAIC regimen significantly prolonged the OS in HCC compared with sorafenib [15]. Ikeda et al. also reported the effectiveness of HAIC in a randomized clinical trial [16]. Therefore, HAIC treatment could be reconsidered for the treatment of HCC. Here, we evaluated the usefulness of New-FP as an initial treatment for advanced HCC using retrospective data; however, this was the largest study to use PSM, which improved the clinical evidence of HAIC, especially within the New-FP for the treatment of HCC.

The presence of MVI or EHS is an important prognostic factor. According to the BCLC staging system, these factors result in a BCLC stage C classification [22]. Although MVI and EHS are both poor prognostic factors, it would be useful to separate them for the analysis of the therapeutic modality. Therefore, we conducted subgroup analyses to clarify the role of New-FP and sorafenib in four cohort groups classified according to the presence of either MVI or EHS. The data before PSM were used for these analyses to prevent the sample sizes of each cohort from becoming too small.

Cohort-1 was the group without MVI and EHS, which is generally categorized as BCLC stage B. The MST of the patients in both groups was relatively longer in cohort-1 than in the other cohorts. The MST of the sorafenib group in cohort-1 was 15 months. The MST of intermediate-stage HCC treated with sorafenib is approximately 12–18 months, which suggests that sorafenib treatment was performed appropriately here [5,35,36]. The OS was not significantly different between the two groups in cohort-1. However, sorafenib treatment was a poor prognostic factor in cohort-1, suggesting that New-FP is a potentially superior initial treatment than sorafenib. As tumor progression degree in the intermediate-stage is broad, further analyses of tumor progression in the intermediate-stage are needed to clarify the benefit of both treatments in cohort-1.

Cohort-2 was the group with MVI but without EHS, which was categorized as BCLC stage C. Thus, this was a locally progressed group without EHS. Although MTAs are recommended in general, the therapeutic outcomes of MTAs for this tumor condition are modest [29]. The MST of patients with MVI-HCC treated with MTAs is only 6–8 months. Kudo et al. revealed that the objective response rate reflected the prolongation of OS [37]. Surprisingly, the objective response rate of New-FP was 71% and the complete response rate of New-FP was 23% according to our previous reports [19,20]. The therapeutic response due to New-FP is the highest compared with that of other reported HAIC regimens, which contributed to the prolongation of OS in cohort-2. Severe MVI was a poor prognostic factor in cohort-2. Kudo et al. reported the effectiveness of HAIC in a randomized phase 3 clinical trial for patients with severe MVI-HCC [17]. The conventional regimen of low-dose FP combined with sorafenib treatment significantly prolonged the survival of patients with severe MVI-HCC compared with sorafenib monotherapy [17]. Since New-FP is a promising treatment for severe MVI-HCC, further analysis will be performed.

Cohort-3 was the group with EHS but without MVI. Only 13 cases were included in the New-FP group because New FP was usually selected for locally progressed cases. The OS was superior in the sorafenib group compared with the New-FP group. Systemic treatment with MTAs is recommended in this cohort. Here, the tumor characteristics of this cohort revealed that tumor diameter was significantly larger in the New-FP than in the sorafenib group, suggesting that the tumor characteristics of the groups in this cohort were substantially different. In cohort-3, further validation of local progression in the hepatic lesions is required.

Cohort-4 was the group with both MVI and EHS which is the most advanced form of HCC. The MST of the patients in cohort-4 highlighted their severely poor prognosis. Since basal prognosis of the patients in cohort-4 is very poor, further development of a treatment is needed for this subgroup. Thus, New-FP is promising as an initial treatment for locally progressed advanced HCC without EHS.

This study had some limitations. First, the study was retrospective. Second, although the initial treatment was divided into New-FP or sorafenib, the choice of post-treatments was on demand; therefore, we cannot deny the possibility of post-treatment. To address this limitation, progression-free survival (PFS) of both groups should be assessed; however, we could not calculate the precise PFS because the study was retrospective and conducted at multi-centers. Third, since the ratio of MVI was relatively high after PSM, the analysis was possibly performed with favorable conditions for the New-FP group. Therefore, we conducted subgroup analyses. Fourth, there are possibilities to occur selection bias because it was a retrospective study. However, here, the several refinements that were described in methods were conducted to avoid selection and post-treatment biases. Therefore, the study results are objective and reliable within the limitations of a retrospective study. To overcome these limitations, we need to perform a randomized prospective study to compare New-FP and MTAs. Recent clinical trial revealed that the combination of atezolizumab plus bevacizumab resulted in better survival and PFS of patients with unresectable HCC than sorafenib [38]. The combination of atezolizumab plus bevacizumab has become the new benchmark for first-line therapy in advanced HCC [39]. Therefore, we need to compare the therapeutic effects of New FP and atezolizumab plus bevacizumab in patients with MVI-HCC. And sequential or combination therapy with New FP and atezolizumab plus bevacizumab might be also promising. Several reports suggest that CDDP or 5-FU used in New FP increases PD-L1 expression in cancer cells [40,41]. High expression of PD-L1 is associated with better therapeutic response to immune-checkpoint inhibitors [42]. Establishment of optimal multidisciplinary therapeutic strategies using locoregional treatments including New FP and the approved systemic therapies is needed in the era of MTA-diversity.

## 5. Conclusions

In conclusion, the current proof-of-concept study showed that local hepatic treatment New-FP therapy prolonged survival than sorafenib in patients with locally progressed HCC. New-FP may be a preferred initial treatment over sorafenib to control local hepatic lesions, which prolongs the survival of patients with HCC.

## Figures and Tables

**Figure 1 cancers-13-00646-f001:**
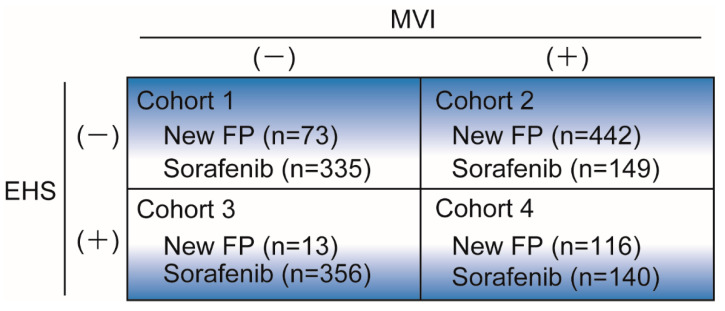
Subgroup analyses.

**Figure 2 cancers-13-00646-f002:**
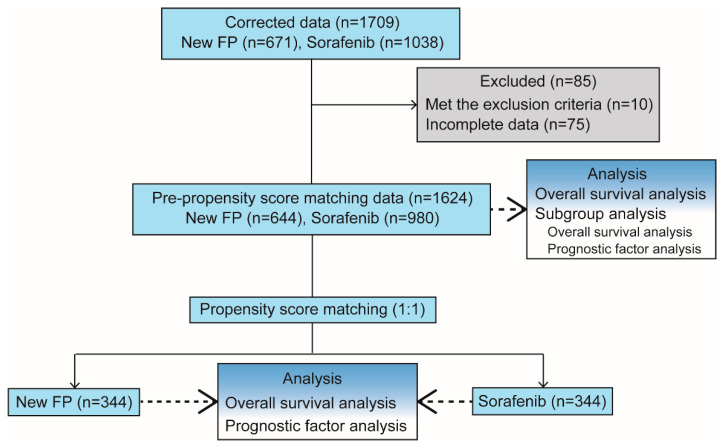
Flow diagram of the study. Abbreviations: MVI, macrovascular invasion; EHS, extrahepatic spread.

**Figure 3 cancers-13-00646-f003:**
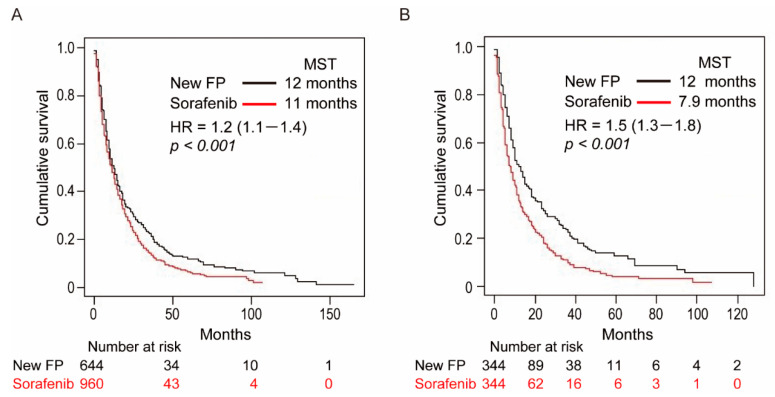
Overall survival curves of patients in the New-FP and sorafenib groups. (**A**) New-FP (*n* = 644, black line) and sorafenib (*n* = 980, red line) before propensity score matching, *p* < 0.001. (**B**) New-FP (*n* = 344, black line) and sorafenib (*n* = 344, red line) after propensity score matching, *p* < 0.001. Abbreviations: MST, median survival time; HR, hazard ratio.

**Figure 4 cancers-13-00646-f004:**
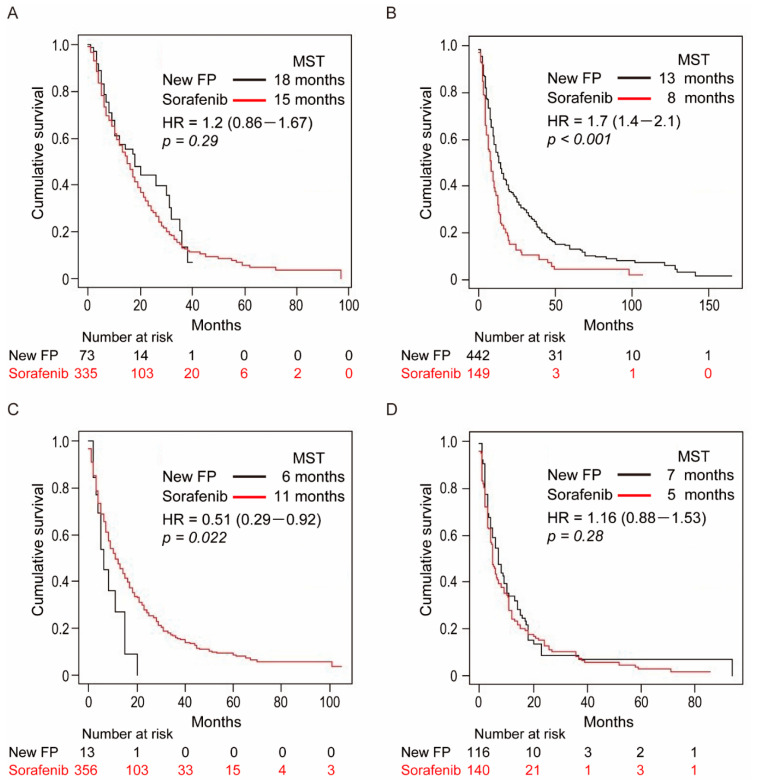
Overall survival curves of patients in the New-FP and sorafenib groups after subgroup analysis. (**A**) Cohort-1 (without MVI and without EHS), New-FP (*n* = 73, black line) and sorafenib (*n* = 335, red line), *p* = 0.29. (**B**) Cohort-2 (with MVI and without EHS), New-FP (*n* = 442, black line) and sorafenib (*n* = 149, red line), *p* < 0.001. (**C**) Cohort-3 (without MVI and with EHS), New-FP (*n* = 13, black line) and sorafenib (*n* = 356, red line), *p* = 0.022. (**D**) Cohort-4 (with MVI and with EHS), New-FP (*n* = 116, black line) and sorafenib (*n* = 140, red line), *p* = 0.28. Abbreviations: MVI, macrovascular invasion; EHS, extrahepatic spread; MST, median survival time; HR, hazard ratio.

**Table 1 cancers-13-00646-t001:** Patient and Tumor Characteristics Before and After Propensity Score Matching.

	Before Matching *n* = 1624			After Matching *n* = 688		
	New-FP *n* = 644	Sorafenib *n* = 980	*p*-Value	New-FP *n* = 344	Sorafenib *n* = 344	*p*-Value
Patient characteristics						
Age (years)	67.94±11.02	70.11±9.52	<0.001	68.50 ± 11.02	68.35 ± 10.33	0.856
Sex Male/Female	505/139	771/209	0.951	267/77	271/73	0.782
HCV	300/344	546/414	<0.001	168/176	152/192	0.252
HBV	123/521	179/801	0.721	71/273	76/268	0.710
Child-Pugh class A/B/C	405/218/21	809/169/2	<0.001	244 /98/2	252/90/2	0.791
Tumor characteristics						
BCLC stage B/C/D	71/571/2	350/630/0	<0.001	70/274/0	89/255/0	0.0857
Tumor size (mm)	110.95 ± 52.83	86.84 ± 60.43	<0.001	109.74 ± 53.42	107.49 ± 54.58	0.585
MVI	558/86	289/691	<0.001	259/85	260/84	1.000
Severe MVI	331/313	168/812	<0.001	129/215	152/192	0.088
EHS	129/515	496/484	<0.001	114/230	122/222	0.574
AFP (ng/mL)	624.20 ± 360.28	609.99 ± 350.87	0.430	598.04 ± 358.41	619.26 ± 342.57	0.428
DCP (mAU/mL)	591.42 ± 346.46	599.33 ± 349.91	0.655	589.06 ± 342.95	612.01 ± 365.21	0.396

Abbreviations: HCV, hepatitis C virus; HBV, hepatitis B virus; BCLC, Barcelona Clinical Liver Cancer; MVI, macrovascular invasion; EHS, extrahepatic spread; AFP, alpha-fetoprotein; DCP, des-γ-carboxy prothrombin.

**Table 2 cancers-13-00646-t002:** Factors Associated With Poor Prognosis After Propensity Score Matching.

Factors	Unit	Hazard Ratio	95% Confidence Interval	*p*-Value
Sorafenib treatment	N/A	1.41	1.19–1.68	<0.001
Presence of severe MVI	N/A	1.53	1.28–1.82	<0.001
Presence of EHS	N/A	1.51	1.26–1.81	<0.001

Abbreviations: MVI, macrovascular invasion; EHS, extrahepatic spread.

## Data Availability

Data is contained within the article or Appendix A.

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
