# Peer review of "Survival Benefit of Hepatic Arterial Infusion Chemotherapy over Sorafenib in the Treatment of Locally Progressed Hepatocellular Carcinoma"

_cancers, 2021, doi:10.3390/cancers13040646_

Round 1
Reviewer 1 Report
The authors present a retrospective analysis of HAI chemo vs sorafenib for HCC. Overall the manuscript is well-written, present valuable information. Number of patients included is high, propensity-score matching (PSM) analysis was performed to limit bias.
Some comments could be made:
Major comments:
- The authors studied prognostic parameters in their 4 cohorts, but did not present the multivariable analysis of the whole cohort before PSM. It should be presented, even if analysis in each cohorts is also very relevant.
- PSM analysis could also be carried in the cohorts with sufficient numbers. Cohort 2 and 4 are particularly relevant.
- Please describe in the methods section the methods used for MVA (criteria for selection in the model, p values, justification of choice for dichotomization of continuous variables…).
- The authors did not discuss at all the new standard of care for advanced HCC, atezolizumab-bevacizumab combination. Please place your work in this context. How HAI wil compare with atezo-bev? Will HAI have a role in second line? Combination with Atezo-bev?
- Some statements are a bit strong against sorafenib and in relation to their treatment. Discussion p 11 “Neither New-FP nor sorafenib could prolong survival, and further development of a treatment is needed.” Sorafenib demonstrated benefit in 2 phase 3 trials. Please amend. Please modify the conclusions about the necessity to perform clinical trials in the era of atezo-bev.
Minor comments:
- Please make statistical analysis in the same direction for MVA for Child-Pugh (A vs B for example, sometimes it goes to B vs A…)
- In the “Refinement to Avoid Bias” section page 5. The authors probably mean « collected » rather than « corrected »?
Reviewer 2 Report
I read with great attention the manuscript entitled " Survival Benefit of Hepatic Arterial Infusion Chemotherapy Over Sorafenib in the Treatment of Locally Progressed Hepatocellular Carcinoma".
The manuscript compares the treatment of hepatocellular carcinoma with Hepatic Arterial Infusion Chemotherapy and sorafenib in terms of survival of the patients.
The following observations are made to provide a better understanding of the study for readers of this journal.
1)The manuscript has some grammatical errors throughout, which loses some clarity due to this. Thorough proof reading is recommended.
2) The authors should give a reference for the propensity score matching method used.
3) Although the authors mention that the patients initially treated with sorafenib were excluded from the New-FP group, was there any other criteria for exclusion?
4) What was the maximum age of the patients?
5) In the point 3.4 “Prognostic Analysis for Survival After PSM” the authors should make it clear that from all the characteristics analyzed only the presence of EHS and severe MVI were independent factors associated with poor prognosis.
6) The authors could add a paragraph regarding the future research.
7) When the authors present the subgroup analysis in the four cohort groups, for each cohort it is presented the factors that showed to be factors that influence the prognostic. The authors could explain the significance of those associations, for example what is the significance that in cohort 2, “C-P class A was a better prognostic factor in this cohort”?
8) The same for the subgroup analysis cohort 3. The authors say that “Hepatitis B virus etiology was a better prognostic factor” but the Hepatitis B virus etiology was not presented has a patient information in the Materials and Methods.
